# Atherogenic Dyslipidemias: Unmet Needs and the Therapeutic Potential of Emerging and Novel Approaches and Drugs

**DOI:** 10.3390/ph16020176

**Published:** 2023-01-24

**Authors:** Alessandra Romandini, Damiano Baldassarre, Stefano Genovese, Stefano Capri, Giulio Pompilio, Marco Scatigna, José Pablo Werba

**Affiliations:** 1Centro Cardiologico Monzino, IRCCS, 20138 Milan, Italy; 2Department of Medical Biotechnology and Translational Medicine, University of Milan, 20133 Milan, Italy; 3School of Economics and Management, Cattaneo-LIUC University, 21053 Castellanza, Varese, Italy; 4Department of Biomedical, Surgical and Dental Sciences, University of Milan, 20122 Milan, Italy; 5Post-Graduate School of Clinical Pharmacology and Toxicology, University of Milan, 20133 Milan, Italy

**Keywords:** atherogenic dyslipidemia, lipid-modifying drugs, cardiovascular disease, drug intolerance

## Abstract

Innovative lipid-modifying agents are valuable resources to improve the control of atherogenic dyslipidemias and reduce the lipid-related residual cardiovascular risk of patients with intolerance or who are not fully responsive to a consolidated standard of care (statins plus ezetimibe). Moreover, some of the upcoming compounds potently affect lipid targets that are thus far considered “unmodifiable”. The present paper is a viewpoint aimed at presenting the incremental metabolic and cardiovascular benefits of the emerging lipid-modulating agents and real-life barriers, hindering their prescription by physicians and their assumption by patients, which need to be worked out for a more diffuse and appropriate drug utilization.

## 1. Introduction

Advances in the knowledge of genetic determinants of lipid disorders and pathways of lipid metabolism led to the discovery of many new potential targets for the pharmacological modulation of atherogenic lipoproteins in blood. In parallel, pharmaceutical companies developed innovative synthetic and biologic compounds directed at inhibiting or stimulating these targets. Some of these compounds are already available on the market, whereas others are in an advanced phase of clinical research. The recent emergence of these novel options for the management of atherogenic dyslipidemias meets the pressing clinical needs to improve lipid control, particularly in patients who still present an unacceptably high lipid-related cardiovascular (CV) residual risk despite treatment with consolidated cholesterol-lowering agents (statins and ezetimibe, hereinafter referred as “standard LDL-cholesterol (LDL-C)-lowering therapy” or “SCLT”). In fact, although a patient’s residual risk is determined by many modifiable non-lipid factors (sedentarism, smoking, obesity, poor blood pressure control, hyperglycemia, etc.), it is widely accepted that suboptimal lipid control may play a key role. The reasons for suboptimal lipid control certainly differ from patient to patient but may end up in a number of clinical problems, as follows:The patient does not tolerate SCLT, due to side-effects, mainly muscle pain, discomfort, or cramps. This leads to scarce adherence to therapy and to the continuous exposure to high levels of atherogenic lipoproteins (Section 3).The patient’s native LDL-C level is very high, such as in some inherited lipid disorders, and/or the patient’s LDL-C goal is very low and the gap between LDL-C achieved on SCLT and the goal remains quite wide (Section 4). The patient has a combined dyslipidemia and shows moderately high levels of triglycerides and low HDL-cholesterol (HDL-C) notwithstanding treatment with SCLT (Section 5).The patient has very high lipoprotein(a) (Lp(a)) levels (alone or with other lipid abnormalities) (Section 6).The patient has a poor adherence to the lipid-modifying treatment (Section 7)

The present paper is a viewpoint aimed at presenting the incremental metabolic and cardiovascular benefits of the emerging lipid-modulating agents and real-life barriers, hindering their prescription by physicians and their assumption by patients, that need to be worked out for a more diffuse and appropriate drug utilization. 

In the interest of clinical readers, relevant features of the novel compounds, summarized in Table 1 and graphically depicted in Figure 1, are discussed not as a drug portfolio but within the frame of the clinical problems that they may help to deal with. It is important to note that this paper is an opinion article, written by a group of researchers who have worked in this field for over 30 years, and not a comprehensive or systematic review or meta-analysis. Herein, we analyze relevant data from the literature and describe personal clinical experiences useful to convey a standpoint, openly recognizing that further publications supporting or confuting our views may have been omitted. Moreover, other authors might interpret the results of the analyzed clinical trials differently and come to different conclusions.

## 2. Methods

The literature selection for this work was carried out through PubMed by searching all the names of the lipid-modifying compounds developed after statins and ezetimibe became the standard of care. Data were extracted from clinical studies describing the effects of the new drugs on lipids levels and on cardiovascular outcomes. Studies regarding lipid-modifying drugs whose clinical research pipeline was interrupted (e.g., torcetrapib) or those withdrawn from the market for any reason (e.g. mipomersen) were not considered.

## 3. Patients with Intolerance to SCLT

A relevant number of patients with a clear-cut indication to treatment with statins are reluctant to take them, even before experiencing any personal adverse events. This may be due to a widespread word of mouth reporting a high risk of “muscle breakdown”. Actually, mild to moderate statin-associated muscle symptoms occur in about 1 out of 10 patients on statins [10], either early or late along the course of the treatment, and these may range from discomfort and weakness to myalgia or myopathy, usually in large muscle groups such as the posterior leg muscles, quadriceps, and gluteal and back muscles. Symptoms typically cease days to weeks after drug interruption. On the other hand, rhabdomyolysis (the literal “muscle breakdown”) is an extremely rare severe event (3.4 affected per 100,000 patients treated per year [11]), generally observed in patients with undetected concomitant triggers such as hypothyroidism or use of interacting drugs. An even rarer side-effect is a statin-induced anti-3-Hydroxy-3-Methylglutaryl-CoA Reductase (HMGCR) myopathy, which is immune-mediated and may require specific treatment [12]. Nevertheless, even mild to moderate muscle symptoms adversely affect the patient’s quality of life and are a substantial determinant of suboptimal statin adherence and treatment discontinuation [13].

Until innovative therapies became an option, expert recommendations to deal with statin-related muscle intolerance included: identifying and eliminating triggers, lowering the statin dose and associating ezetimibe to maintain LDL-C-lowering potency, trying a different posology statin scheme at the expense of efficacy (e.g., five days a week, every other day, twice a week, and once a week) or switching to a differently metabolized statin [14]. Though straightforwardly shifting to new LDL-C-lowering compounds with better tolerability is nowadays often requested by patients and may seem suitable, we believe that the evidence of the CV benefits produced by statins warrants the application of any possible strategy to maintain a statin, at least at a low dose, as part of the lipid-lowering regime [15]. However, if LDL-C levels remain quite far from the individual target or the patient is fully statin-intolerant, bempedoic acid, anti-proprotein convertase subtilisin-kexin type 9 (PCSK-9) monoclonal antibodies (mAbs; namely evolocumab and alirocumab), or small interfering ribonucleic acid (siRNA) against PCSK-9 mRNA (namely inclisiran) may be considered. 

Bempedoic acid (BA) is an inhibitor of ATP-citrate lyase, an enzyme upstream in the pathway of cholesterol synthesis. In addition, BA is an oral prodrug, activated by the enzyme very-long-chain acyl-CoA synthease-1 (ACSVL1), which has a high expression in the liver but is undetectable in skeletal muscle. It is reported that, thanks to this mechanism, BA does not lead to clinically meaningful muscle-related symptoms and is, therefore, depicted as a kind of “muscle-sparing” statin equivalent [16].

BA is approved in the USA and European Union (EU) as an adjuvant to maximally tolerated statin therapy in patients with atherosclerotic CV disease and patients with heterozygous familial hypercholesterolemia (HeFH). Though the mean incremental LDL-C-lowering effect of cholesterol synthesis pathway serial inhibition (upstream with BA, downstream with a statin) is small (about 18% [17]), adding BA may be enough to approach or even reach the LDL-C target in patients with LDL-C levels very near their target on the highest tolerated doses of background SCLT [18,19]. Clinicians should be aware that when BA is administered in association with statins, blood levels of the latter increase 1.5- to 2.0-fold, which, in turn, might increase the probability of statin-related side-effects. Accordingly, the summary of product characteristics indicates that BA should be added to simvastatin at doses no higher than 40 mg/day [20], and we also deem it reasonable to avoid using maximal doses of other statins when combined with BA.

Moreover, BA is also approved in the EU to treat fully statin-intolerant patients (unable to take any dose of any statin), or patients in which statins are contraindicated (e.g., patients with primary myopathies). As BA monotherapy has only a modest LDL-C-lowering effect of about 12.0–22.9% [21], the ideal fully statin-intolerant candidate for BA monotherapy is the patient with only a small gap between untreated LDL-C and their individual goal. Alternatively, in patients with an LDL-C gap between 30 and 40%, an attempt with the fixed-dose combination of BA plus ezetimibe (or adding a pill of ezetimibe to BA), reported to reduce mean LDL-C levels by 39.2% [22], seems warranted. BA should be indicated with caution in patients with a history of hyperuricemia and/or gout (as it may increase uric acid levels and induce gout attacks) [23]. It is worth noting that the clinical benefits of BA on hard CV endpoints have not been proven yet. In this regard, the results from the CLEAR Outcomes trial [1] are planned to be reported soon.

Alirocumab and evolocumab are mAbs against circulating PCSK-9. These compounds, available in the LDL-C-lowering portfolio since 2015 [24,25], are administered subcutaneously every 14 days (75 or 150 mg of alirocumab; 140 mg of evolocumab) or 28 days (420 mg of evolocumab). They have an excellent safety profile and, in patients on background statin therapy, they produce intense mean LDL-C reductions of about 60% [26]. However, PCSK-9 inhibition may be partly compensated by a combined slightly increased cholesterol synthesis and absorption [27] and, consequently, in patients with full intolerance to SCLT, the observed mean response to anti-PCSK-9 mAbs alone is usually somewhat less impressive, with an approximate 45–53% LDL-C reduction [28,29].

When, in a fully statin-intolerant patient, the LDL-C levels achieved with PCSK-9 mAb treatment alone or combined with ezetimibe remain far distant from the individual target, it might be reasonable to test the addition of BA, which seems to produce an incremental LDL-C reduction of about 30%, as reported in a short-term study [30]. 

Inclisiran is a double-stranded siRNA that drives the catalytic breakdown of the mRNA coding for PCSK-9, preventing its translation into its protein [31]. This increases LDL-C receptor recycling and expression on the hepatocyte cell surface, which increases LDL-C uptake and lowers blood LDL-C levels. Inclisiran is conjugated to triantennary N-acetylgalactosamine (GalNAc), which binds to highly liver-expressed asialoglycoprotein receptors, leading to the uptake of inclisiran primarily into hepatocytes [32,33]. In other words, whereas mAbs against PCSK-9 neutralize circulating PCSK-9, inclisiran directly blocks PCSK-9 synthesis in liver cells. Another main difference between the mAbs and the siRNA is the longer-term action of the latter. Accordingly, the recommended dose of inclisiran (284 mg) must be administered subcutaneously at the start, after 3 months, and henceforth every 6 months.

While the mean LDL-C reduction gained by adding inclisiran is about 52% in patients on SCLT [34], in those treated with inclisiran alone, the effect is, as described above for mAbs, somewhat smaller (38–42%) [35].

Importantly, the smaller relative LDL-C reduction obtained with mAbs or inclisiran therapy alone should not be considered as a problem or discourage their use. Actually, patients fully intolerant to SCLT commonly show higher baseline LDL-C levels than those on SCLT; therefore, their absolute LDL-C reduction and the corresponding CV risk reduction with mAbs or inclisiran alone may be even larger than in patients on SCLT [36]. For example, considering two patients at high baseline CV risk with similar clinical characteristics other than lipid levels, a 40% LDL-C-lowering starting from an LDL-C of 180 mg/dL yields a larger absolute reduction in the risk of CV events than a 60% LDL-C-lowering starting from an LDL-C of 80 mg/dL, even though the LDL-C level in the former remains quite far from their currently recommended LDL-C goal [15]. These numbers may be relevant for individual cost–benefit considerations, and become more evident by comparing the estimated number needed to treat (NNT) to prevent a primary event in the whole cohort of ODYSSEY Outcomes (NNT = 62.5) with that of the study subgroup of patients without statins (NNT = 12.5) (data computed from the study supplemental material) [3]. As for inclisiran, cost–benefit considerations may be thus far just a theoretical estimate of CV risk reduction based on the extent of LDL-C reduction [37], as the actual effect of this innovative drug on CV outcomes is still unknown (the results of the ongoing ORION-4 phase 3 trial are expected to be reported in 2026) [4]. 

Another major potential adverse effect of LDL-lowering therapy that may naturally worry both patients and clinicians is the increased risk of developing diabetes. In mendelian randomization studies, inherited variants in the genes encoding HMGCR (the target enzyme of statins) and PCSK-9 are both associated with an approximate 10% increased risk of diabetes for each 10 mg/dL decrease in LDL-C [38]. Yet, whereas genetic variants may predispose to diabetes by affecting gene expression in different tissues and organs, pharmacological compounds modulate gene expression specifically in cells and tissues where they are distributed. Indeed, the 10–12% increased risk of new-onset diabetes mellitus occurring in patients treated with statins [39,40], which are widely distributed to different organs and tissues, was not observed either with mAbs against PCSK-9 [2,41], which neutralize circulating PCSK-9 without entering cells, or with inclisiran, presumably due to its high siRNA-GalNAc conjugate hepatotropism [33].

Interestingly, several placebo-controlled studies (longest follow-up 1 year) and a metanalysis reported that, contrarily to statins, bempedoic acid has a neutral (or even preventive) effect on the development of new-onset diabetes mellitus [42,43,44], suggesting that the diabetogenic effect of statins is not related to the inhibition of cholesterol synthesis in the liver but it may be associated with other yet uncovered mechanisms.

Based on these data, we believe that the better tolerability profile of the new LDL-C-lowering drugs, compared to SCLT, will substantially lower the occurrence of side-effects as a main determinant of a reduced treatment adherence and persistence. 

## 4. Patients with Severe Primary Hypercholesterolemia or with Very Ambitious LDL-C Goals

Familial hypercholesterolemia (FH) due to a pathogenic mutation in one of the five thus far identified genes that take part in cholesterol metabolism (LDL Receptor (LDLR), apolipoprotein B (APOB), PCSK-9, apolipoprotein E (APOE), signal transducing adaptor family member 1 (STAP1), and LDL receptor adaptor protein 1 (LDLRAP1)) is a monogenic disease and is probably the most harmful form of hypercholesterolemia. This is particularly evident in the rare homozygous patients (Homozygous Familial Hypercholesterolemia, HoFH) [45], but it is also true for the more common heterozygous patients (HeFH) [46]. In fact, the exposure to high levels of atherogenic lipoproteins from birth conveys an exceptionally high CV risk [47] and the strong recommendation is to reduce LDL-C levels as early and intensively as possible [15,48]. 

Until recently, the alternatives to control hypercholesterolemia in patients with HoFH were very limited and invasive. In fact, these patients had to undergo, on top of SCLT, LDL-apheresis weekly or every other week or, as a last resource, liver transplantation [45]. These therapies are still valid options for contexts where the innovative PCSK-9 inhibitors, lomitapide and/or evinacumab, are not available or cannot be afforded [49].

Evolocumab is the only PCSK-9 inhibitor so far approved for use in patients with HoFH [25]. Though the mean LDL-C reduction obtained with evolocumab is modest in these patients (about 20% at 12 weeks), the individual response is highly variable and is strongly influenced by the type of mutations inherited, reaching up to 90% in some cases [50]. Therefore, given its excellent safety profile and relatively low cost compared to the alternatives (see below), unless the patient is known to have null mutations in both alleles, a 12-week cycle with evolocumab is, in our view, the first therapeutic approach that should be tried since age 10, on top of SCLT, in these unfortunate young patients. A very recent phase 3 study showed a comparable LDL-C response variability with alirocumab [51].

Yet, if LDL-C levels remain unacceptably high, then the “orphan drugs” lomitapide or evinacumab may be considered, usually associated with background SCLT and/or PCSK-9 inhibitors. With each of these compounds, a further 50% mean LDL-C reduction may be obtained (with a wide inter-individual variability), through mechanisms that are independent of the LDLR functionality. Lomitapide is an orally administered synthetic inhibitor of the liver and gut enzyme microsomal transfer protein (MTP), which reduces lipoprotein assembly and secretion [52], whereas evinacumab is an intravenous administered mAb, which promotes very-low-density lipoprotein (VLDL) clearance upstream of LDL formation by targeting a protein named angiopoietin-like protein 3 (ANGPTL3) [53].

While the therapeutic choice for patients with HoFH indeed requires the expertise of a lipid specialist, this is formally mandatory for lomitapide and evinacumab, not only for the need to strictly monitor safety issues, but also for their extremely high cost.

On the other hand, the management of patients with HeFH is radically different. Given the relatively high prevalence of this genetic disease in the general population (about 1:250 to 1:500) [54], pediatricians, general physicians, and the different specialists involved in CV prevention (cardiologist, diabetologist, endocrinologist, nephrologist, etc.), aside from the lipidologist, commonly need to face patients with this disorder, which may pose some difficulties. Specifically, though LDL-C in most patients with HeFH may be suitably well controlled with SCLT, some of them show very high baseline LDL-C levels (typically > 220–240 mg/dL), which implies the need of a large treatment effect to reach (or at least to approach) the ambitiously low LDL-C goals recommended by current guidelines [15]. Moreover, in patients with HeFH in primary prevention, generally young and often sportive, statins may increase the incidence of exercise-related muscle complaints and augment the exercise-induced rise in muscle enzymes [55]. 

The PCSK-9 inhibitors are particularly indicated for severely hypercholesterolemic patients hardly controlled with SCLT. Numerous studies show that the efficacy of PCSK-9 inhibitors in HeFH is comparable to that observed in patients with polygenic hypercholesterolemia (LDL-C reduction of 50–60% for mAbs; 40–50% reduction for inclisiran) [54,56,57,58]. It is worth noting that only evolocumab received authorization for use in children with HeFH from 10 years of age. In Italy, alirocumab, evolocumab, and inclisiran are fully reimbursed by the National Health System in patients in primary prevention with HeFH if LDL-C remains ≥ 130 mg/dL notwithstanding SCLT at the maximal tolerated doses [59]. Other European countries have more stringent criteria for reimbursement [60].

On the other hand, the ambitiously low LDL-C goals recommended by expert guidelines for patients at very high CV risk are often not achieved with statins alone. Indeed, beyond the problems of tolerability discussed above, the individual response to statins is highly variable and some patients are low-responders [61]. In many cases, the problem might be solved by adding ezetimibe, which provides an incremental reduction in LDL-C levels of about 20% [62]. Yet, even using maximally tolerated doses of high-potency statins plus ezetimibe, a significant LDL-C gap may still remain, and, in these patients, PCSK-9 inhibitors may be added to the scheme [63]. Although for LDL-C levels, the notion “lower is better” is already undisputable [64] and the high LDL-C-lowering potency and good safety profile of PCSK-9 inhibitors are acknowledged [65,66,67], the debatable issue is, given their high current cost, which are the patient’s features in which the use of these compounds is cost/effective. This facet of the clinical assessment might be pointless in contexts with ample resources but becomes relevant in economically stressed public health systems. Some hints about cost-effectiveness may be gained from subgroup analyses of the CV outcome trials with these drugs [2,68,69,70,71] and from some informative metanalyses [36]. All in all, these sources suggest a better cost/benefit ratio and, therefore, a priority use in patients with (a) LDL-C levels > 100 mg/dL before adding the PCSK-9 inhibitor [3]; (b) a recent acute myocardial infarction or multiple prior myocardial infarctions or residual multivessel coronary artery disease [69,70]; (c) clinical atherosclerosis in multiple vascular beds [72]; (d) multiple metabolic risk factors [71].

Promising news recently came from the phase 2 ROSE study with obicetrapib [73], a novel synthetic oral inhibitor of the cholesteryl ester transfer protein (CETP). Although obicetrapib belongs to a drug family that thus far failed in phase 3 trials to demonstrate a cardioprotective effect notwithstanding remarkable HDL-C-raising actions, this new compound also has an LDL-C-lowering potency close to PCSK-9 inhibitors, as well as other favorable lipid effects (see below). Therefore, if the ongoing phase 3 PREVAIL study [7] demonstrates that obicetrapib not only induces relevant lipid changes but also meaningfully reduces CV residual risk, this drug might turn to be a hopefully less costly alternative to PCSK-9 inhibitors.

All things considered, our view is that in very-high-risk patients and in patients with HeFH with LDL-C levels unlikely to reach their goal with a high-potency statin alone, it is advisable to start a straightforward statin-ezetimibe combination. If LDL-C levels remain unacceptably high, given the wide interindividual variability of the LDL-C response to bempedoic acid [17], we believe that a short-term test with this drug on top of the background SCLT before prescribing a life-long, more costly treatment with a PCSK-9 inhibitor may be a reasonable approach, unless the gap with the LDL-C target is extremely large. 

## 5. Patients with Mixed Dyslipidaemia

Combined (or mixed) dyslipidaemia is the occurrence of hypercholesterolemia in concert with hypertriglyceridemia and low HDL-C [74]. This disorder is generally associated with other dysmetabolic features such as insulin resistance and central obesity (often referred to as metabolic syndrome) or diabetes mellitus. Though moderately high triglycerides and low HDL-C are both epidemiologically associated with an increased CV risk [75,76], most available evidence suggests that these deviations are markers of concurrent lipid abnormalities (i.e., high levels of intermediate-density lipoprotein (IDL), small dense LDL, and APOB), which are more probably the actual atherogenic factors [77,78,79] and potential targets of intervention. 

Among the currently available lipid-modifying drugs (leaving behind the old nicotinic acid derivatives), two types of compounds may be considered to control moderately high triglycerides: fibrates and omega-3 fatty acids. Yet, in patients with well-controlled LDL-C, whether the addition of these compounds produces only slight favorable changes in the patients’ laboratory results or actually reduce their residual CV risk is still a matter of controversy, even among expert lipidologists. 

To make the long fibrates’ story short, while early clinical studies showed that these potent triglyceride-lowering compounds significantly prevent first and recurrent CV events in patients without background SCLT [80,81], they were indeed ineffective in terms of primary outcomes reduction in trials with patients ON background SCLT [82], which is the current standard of care [15]. Post hoc analyses of these trials showing some positive results (i.e., reduced major CV events but no decrease in CV or total mortality) [83,84,85,86,87], specifically in patients with high triglycerides and low HDL-C at baseline, left for many years the notion that the addition of fibrates to SCLT could reduce the residual risk of patients with combined dyslipidaemia, though confirmation was needed. Disappointingly, the widely expected results of the PROMINENT study, a CV outcome trial (CVOT) with a new fibrate (pemafibrate) in patients with high CV risk and mild to moderate hypertriglyceridemia, were negative [88], making it even more difficult for fibrates to still retain a place in lipid therapy. In fact, whether inefficacy was related specifically to pemafibrate or may be generalized to older fibrates will certainly be the subject of theoretical analyses; yet, further trials with off-patent fibrates to corroborate or confute PROMINENT are realistically unexpected.

Even more contradictory is the current knowledge about the putative cardioprotection endowed by the omega-3 fatty acids eicosapentaenoic acid (EPA) and docosahexaenoic acid (DHA). These compounds produce a modest dose-dependent reduction in triglycerides levels, which is larger in subjects who have higher baseline levels, without changing or slightly increasing LDL-C [89]. Whereas borderline positive results of an early open-labeled CVOT with low doses (1 g/day) of a mix of EPA and DHA in patients with a recent acute myocardial infarction [90] prompted the medical community to a diffuse prescription for secondary prevention purposes, a subsequent systematic Cochrane review of randomized clinical trials indicated little or no effect of omega-3 supplements on mortality or CV health [91]. This new information led the EMA to stop the recommendation. More recent studies with newer omega-3 fatty acids formulations did not provide consistent responses to the matter. Indeed, the initially exciting results of REDUCE-IT (a 25% reduction in the primary CV outcome with 4 grams/day of icosapent ethyl in statin-treated patients with high triglyceride levels) [5] might be at least in part overestimated by the use of a not-inert placebo (mineral oil), which had several mild adverse metabolic effects (increased APOB, LDL-C, non-HDL-C, and high-sensitivity PCR (hsPCR)). Moreover, two other contemporary CVOTs with omega-3 fatty acids, the placebo-controlled STRENGTH (4 grams/day of a carboxylic acid formulation of EPA and DHA vs. corn oil [92]) and the open-label RESPECT-EPA (1.8 grams/day of icosapent ethyl vs. usual care) [6], failed to achieve a significant reduction in the primary endpoint. It is worth noting that in the three trials, there was a significant increase in the incidence of atrial fibrillation (AF), with a number needed to harm (NNH) equal to 71 (REDUCE IT) [5], 114 (STRENGTH) [92], and 67 (RESPECT-EPA) [6], indicating that both EPA-DHA formulations and icosapent ethyl may induce AF. Therefore, whatever the reasons for the discrepancy between the results of these studies are (not-inert placebo, EPA/DHA mixture vs. pure EPA, study design, baseline triglycerides, EPA dose, baseline EPA levels, etc.), a reasonable uncertainty about the usefulness of these compounds is still there, whereas their safety profile warrants caution. We, thus, believe that, for the time being, further research is needed to identify which patients most likely will benefit from high doses of icosapent ethyl and which are likely to harm.

The bottom line of this section is that the pharmacological treatment gap to reduce residual risk in patients on maximal SCLT who remain with combined dyslipidaemia is still unfilled. Although the experimental CETP inhibitor obicetrapib—previously mentioned for its potent LDL-C-lowering efficacy—reduces triglyceride levels only modestly (about 11%), at the tested doses of 5 and 10 mg/day, it favorably modifies non-HDL-C (−39%; −44%), HDL-C (+135%; +165%), APOB (−24%; −30%), and Lp(a) (−34%; −56%), suggesting a potential role for this compound in reducing the residual risk of patients with combined lipid disorders. Another promising compound to treat this category of patients is olezarsen, a GalNAc-conjugated antisense oligonucleotide targeted to hepatic apolipoprotein C3 (APOC3) mRNA to inhibit APOC3 protein production. In fact, a recent study showed that, at the different doses tested administered subcutaneously in patients on background LDL-C-lowering therapy with moderate hypertriglyceridemia, olezarsen reduces triglycerides by 23 to 60%, VLDL-C by 27 to 58%, and APOB up to 17%, while it increases HDL-C by 11 to 40%. Several years will pass before we know what the effect of these innovative drugs on CV outcomes is, and whether they may be used to fill this gap.

## 6. Patients with Hyperlipoprotein(a)

Epidemiology has consistently demonstrated an association between high Lp(a) levels and risk of atherosclerotic disease in different vascular districts [93,94], and these findings were recently corroborated in mendelian randomization studies [95,96]. Lp(a) in the general population is primarily genetically determined (70 to 90% of the interindividual variation) [97,98], though impairment of renal function may also lead to “acquired” hyperlipoprotein(a)emia [99]. Whereas 30 mg/dL is conventionally considered the upper limit of a normal plasma Lp(a) value, what a clinically meaningful high Lp(a) level is, is not yet well established. What seems clear is that there is a positive linear association between Lp(a) levels and CV risk [100,101], and that values higher than 180–200 mg/dL may determine a risk such as the one endowed by HeFH [15,96]. Importantly, hyperlipoprotein(a)emia might turn out to be a target for the personalized reduction in the residual risk that remains in patients in secondary prevention notwithstanding a good control of traditional risk factors [94,102,103]. In a recent large epidemiological Danish study, the authors estimated that to achieve a 20% and 40% risk reduction of major adverse coronary events (MACEs) in secondary prevention, plasma Lp(a) levels should be lowered by 50 mg/dL and 99 mg/dL, respectively, for 5 years [104]. Such an intense effect could be obtained thus far only acutely by using lipoprotein apheresis, but the mean interval reductions with weekly or biweekly procedures reach around 25–40% [105]. In fact, Lp(a) levels are not reduced by SCLT and may even be slightly increased (10–20%) by statins [106], whereas the Lp(a)-lowering effect of PCSK-9 inhibitors is modest with a 20–30% reduction [107,108]. Specific Lp(a)-lowering drugs that target the production of apolipoprotein(a) were recently developed, and some of them are at an advanced stage in clinical research assessment. Pelacarsen is an antisense oligonucleotide (ASO) conjugated with GalNAc3 that promotes the degradation of apolipoprotein a (APO (a)) mRNA in liver cells, reducing the synthesis of APO(a) [109]. The compound is administered subcutaneously every month. In phase 1 and 2 clinical studies, pelacarsen reduced Lp(a) levels by 35–80%. Safety and tolerability are good, but a transient flu-like syndrome occurs in some patients within the first 24 hours after each dose. HORIZON is an ongoing phase 3 placebo-controlled study to investigate the efficacy of pelacarsen in reducing the recurrence of CV events [8]. The results of HORIZON are expected in year 2026. 

Olpasiran is a siRNA conjugated with GalNAc3 that inhibits the translation of APO(a) mRNA in liver cells by a complex molecular mechanism, thus reducing the synthesis of APO(a). The compound is administered subcutaneously every 3 months. In phase 1 and 2 clinical studies, olpasiran reduced Lp(a) levels by 67 to 97% according to the dose [110,111]. Interestingly, at the highest dose tested (225 mg), the extent of Lp(a)-lowering was similar when olpasiran was administered every 12 or every 24 weeks (both −97%), demonstrating the long persistence of its pharmacological action. A phase 3 CV outcomes trial with olpasiran (namely OCEAN(a)) is planned to start soon [9].

The results of CVOT with these lipoprotein(a)-lowering compounds will allow for confirming or confuting the “lipoprotein(a) hypothesis”, hopefully mirroring the LDL-C hypothesis history. Until then, knowing a patient’s Lp(a) level may be useful as a risk marker to modulate interventions on modifiable consolidated risk factors, as recommended by the latest lipid guidelines [15]. 

## 7. Poor Compliance to the Lipid-Modifying Treatment

Poor adherence to lipid-modifying agents is a frequent clinical problem that may worsen the patients’ prognosis [112]. This phenomenon is more pronounced in primary prevention patients than in secondary prevention patients, but it remains a major obstacle even in the latter, who show a progressive loss of adherence with the passage of time after an acute vascular event [113]. Some new lipid-modifying drugs may help to reduce factors associated with a low compliance: (a) “too many pills” is a frequent claim for reduced compliance; utilization of fixed-dose drug combinations (e.g., statin-ezetimibe or BA-ezetimibe) aids to reduce the high pill burden that patients with CV disease often assume; (b) “muscle pain or cramps” with SCLT, another common reason for low adherence, may be faced by shifting to the emergent LDL-C-lowering compounds BA and/or PCSK-9 inhibitors, as described above (paragraph 2); (c) “forgetfulness to take the pills for cholesterol every day”, a problem commonly referred by busy active workers and by elderly people, may be tackled by using pill organizers, available electronic pill reminders (apps), or, if clinically indicated, by shifting to the less frequently administrated anti-PCSK-9 mAbs or siRNA.

## 8. Real-Life Barriers for Using Innovative LDL-C-Lowering Drugs

The advent of new resources to treat lipid disorders more efficiently and comprehensively came not without some significant obstacles. One of these barriers is a physicians’ inertia to modify consolidated prescription habits and/or to adopt recommendations of updated expert guidelines. Evidence that this problem is not limited to the field of dyslipidemias comes from the results of a recent real-world Italian study showing that, among diabetics with coronary artery disease, only 14.3% were treated with the proper GLP-1 or SGLT2-i, whereas 16.5% received sulphonylureas, which guidelines contraindicate in these types of patients [114]. Similarly, notwithstanding adding ezetimibe to a background statin therapy to reach lower LDL-C levels in patients at high or very high risk and already a class IIa, level of evidence B recommendation in the 2016 ESC/EAS guidelines for the management of dyslipidemias [115], it is still barely utilized (9%) [116], even though it is available as a generic drug, it has a low cost, and a number of combined statin-ezetimibe formulations are available on the market. This barrier might be overcome by implementing in the next lipid guidelines the agreeable proposal of an expert group: “using a statin-ezetimibe combination upfront in very-high-risk patients with high LDL-C unlikely to reach goal with a statin, and in primary prevention familial hypercholesterolemia patients” [117]. 

Another barrier is prescription intricacy. Without considering lomitapide and evinacumab, whose use is restricted to expert lipidologists, evolocumab, alirocumab, and inclisiran may be prescribed by other specialists. In Italy, for example, prescription is granted only to internists, cardiologists, endocrinologists, and diabetologists, and the formal procedure that these physicians are required to fulfil to grant the patient public reimbursement of the treatment and to refill prescriptions is troublesome and time-consuming; therefore, each patient included in a life-time treatment with a PCSK-9 inhibitor entails a growing and hardly bearable administrative workload. Simplifying formalities and allowing a progressive extension of the coverage period of each refilling prescription might be of significant help. Alternatively, patients who need treatment with these drugs might converge to lipid clinics where ad hoc systems to ease administrative workload might be implemented. 

Finally, though the potent LDL-C-lowering effect and good safety profile of PCSK-9 inhibitors invites to foresee a much larger use of these compounds, it is currently restricted by their high cost. If stakeholders will find strategies to market these products at lower prices, a wider range of patients will have the chance to benefit. In any case, the future for clinical lipidology and for patients affected by atherogenic dyslipidemias seems bright, as synthetic compounds with analogous lipid-modifying efficacy and, hopefully, more affordable costs are in the pipeline of several pharmaceutical companies.

## 9. Conclusions

For several decades, physicians have had limited pharmacological options to manage patients with atherogenic dyslipidemias, and several therapeutic needs were not being met. Innovative lipid-modifying compounds already licensed (evolocumab, alirocumab, inclisiran, bempedoic acid, lomitapide, and evinacumab) can solve clinical problems such as drug intolerance, lipid-lowering in extremely severe primary hypercholesterolemia, or achievement of very low LDL-C targets. Unfortunately, thus far, their use is limited by a number of significant cultural, organizational, and economic barriers that still need to be overcome. In addition, the rapid pace of drug development in this therapeutic area and the contrasting results of some clinical trials (e.g., with different omega-3 fatty acids) is challenging scientists and clinicians to identify the correct indications and appropriate patient for each agent. Finally, new compounds at an advanced stage of research (e.g., pelacarsen, olpasiran, obicetrapib, and olezarsen) will hopefully allow the control of atherogenic lipid abnormalities heretofore considered unmodifiable and to further reduce the residual lipid-related CV risk of our patients.

## Figures and Tables

**Figure 1 pharmaceuticals-16-00176-f001:**
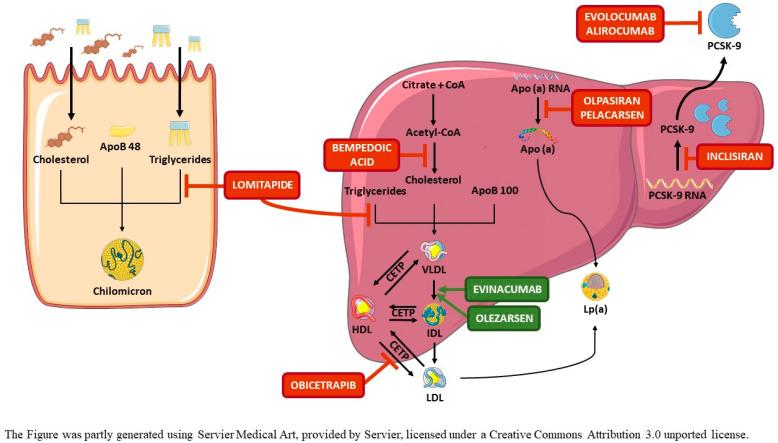
Mechanisms of action of emerging lipid-modifying drugs. BA reduces cholesterol synthesis in cells by inhibiting ATP-citrate lyase. BA is a prodrug activated by ACSVL1, which is selectively expressed in hepatocytes. Activated BA, similarly to statins, upregulate the expression of the LDLR to compensate for reduced intracellular cholesterol. Evolocumab and alirocumab neutralize circulating PCSK-9, whereas inclisiran binds to PCSK9 mRNA, mainly in liver cells, inducing its degradation and, thus, the synthesis of PCSK-9. In both cases, loss of PCSK-9 function reduces the degradation of LDLR, increasing their expression in cell membranes. Lomitapide inhibits MTP, which reduces the assembly between lipids and apoB in hepatocytes and enterocytes and, thus, the production and secretion of VLDL and chylomicrons. Evinacumab neutralizes circulating ANGPTL3, thus reducing its inhibitory effect on LPL and EL, leading to an increased lipolysis of VLDL and clearance of IDL, upstream of LDL formation, by a putative hepatic IDL-receptor. Obicetrapib inhibits CETP, which mediates the transfer of cholesteryl esters from HDL to the APOB-containing VLDL-IDL-LDL, markedly raising HDL-C and lowering cholesterol contained in atherogenic particles. Olezarsen reduces APOC3 production by liver cells by inhibiting APOC3 mRNA transcription. Reduction in APOC3 function releases LPL from its inhibitory effect, enhancing the lipolytic cascade. Pelarcansen promotes the degradation of APO(a) mRNA in liver cells while Olpasiran blocks the translation of APO(a) mRNA in liver cells, leading, in both cases, to a reduced synthesis and secretion of lipoprotein(a).
Abbreviations: ACSVL1: very-long-chain acyl-CoA sinthetase-1; ANGPTL3: angiopoietin-like protein 3; APO(a): apolipoprotein (a); APOB: apolipoprotein B; APOC3: apolipoprotein C3; BA: bempedoic acid; CETP: cholesteryl ester transfer protein; EL: endothelial lipase; HDL: high-density lipoproteins; HDL-C: cholesterol contained in HDL; IDL: intermediate-density lipoproteins; LDL: low-density lipoproteins; LDLR: LDL-receptor; Lp(a): lipoprotein(a); LPL: lipoprotein lipase; MTP: microsomal transfer protein; PCSK-9: proprotein convertase subtilisin/kexin type 9; TG: triglycerides; VLDL: very-low-density lipoproteins.

**Table 1 pharmaceuticals-16-00176-t001:** Relevant features of the new lipid-lowering compounds.

Drugs	Main Lipid Target(s)	Mechanism of Action	Route of Administration	Clinical Outcomes Evidence of CV Benefit (Citation Number)	On the EU Market
Bempedoic acid	LDL-C	Inhibition of cholesterol synthesis	oral	CLEAR Outcomes [1]. Results expected in 2023	Yes
Evolocumab	LDL-C	Neutralization of PCSK-9 in blood	subcutaneous	FOURIER [2]	Yes
Alirocumab	LDL-C	Neutralization of PCSK-9 in blood	subcutaneous	ODYSSEY OUTCOMES [3]	Yes
Inclisiran	LDL-C	Inhibition of PCSK-9 synthesis	subcutaneous	ORION-4. Results expected in 2026 [4]	Yes
Lomitapide	LDL-C	Inhibition of MTP-mediated lipoprotein assembly	oral	Not announced	Yes
Evinacumab	LDL-C	Inhibition of ANGPLT3 and increased VLDL clearance	intravenous	Not announced	Yes
Icosapent ethyl	TG	Not clear	oral	REDUCE-IT [5], RESPECT-EPA [6]	Yes
Obicetrapib	LDL-C, non-HDL-C, HDL-C, Lp(a)	Inhibition of CETP	oral	PREVAIL [7]. Results expected in 2026	No
Olezarsen	TG, non-HDL-C, HDL-C	Inhibition of apoC-3 synthesis	subcutaneous	Not announced	No
Pelacarsen	Lp(a)	Inhibition of apo(a) synthesis	subcutaneous	HORIZON [8]. Results expected in 2025	No
Olpasiran	Lp(a)	Inhibition of apo(a) synthesis	subcutaneous	OCEAN(a) [9]. Results expected in 2026	No

Abbreviations: CETP: cholesteryl ester transfer protein; HDL-C: high-density lipoprotein-cholesterol; IV: intravenous; LDL-C: low-density lipoprotein-cholesterol; Lp(a): lipoprotein(a); N/A: not available; PCSK-9: proprotein convertase subtilisin/kexin type 9; TG: triglycerides; VLDL: very-low-density lipoprotein.

## Data Availability

Data sharing not applicable.

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
