# Peer review of "Atherogenic Dyslipidemias: Unmet Needs and the Therapeutic Potential of Emerging and Novel Approaches and Drugs"

_pharmaceuticals, 2023, doi:10.3390/ph16020176_

Round 1

Reviewer 1 Report

Dear Editor,

I carefully read the manuscript "Atherogenic dyslipidaemias: unmet needs and how the new available or upcoming drugs might fulfill them" by Romandini et al.

My comments and suggestions for the authors are the following:

 - English language needs to be revised and improved. 

 - Line 41: Among the main reasons for suboptimal lipid control, the authors should also mention a bad compliance to pharmacological treatment.

 - Throughout the manuscript, not all of the authors' claims are supported by references. For example, a reference should be added at line 106, 109, 110, ...

 - In the manuscript, all abbreviations should be specified at their first occurrence.

 - In the manuscript, the authors should also refer to findings from meta-analyses summarizing evidence in the field. In particular, they should refer to doi: 10.1371/journal.pmed.1003121, doi: 10.1007/s40264-020-00931-6, doi: 10.1161/JAHA.122.025551, doi: 10.1097/MD.0000000000031199, doi:10.1016/j.ahjo.2022.100127, doi: 10.1016/j.atherosclerosis.2020.09.021. 

 - Table 1 - What about mipomersen?

 - Table 1 - References should be added in the table.

 - In the manuscript, the authors should include a "methods" section detailing how literature searching process was performed.

 - References are not reported in accordance with the Instructions for the Authors.

 - In the Conclusions, the authors should specify that this is a state of the art review (and not a systematic review) and even the other limitations associated to their not systematically approach. In effect, I got the impression that the authors only reported the information and sources that supported their thesis. Readers should be properly informed of this.

Author Response

REVIEWER 1

Dear Editor,

I carefully read the manuscript "Atherogenic dyslipidaemias: unmet needs and how the new available or upcoming drugs might fulfill them" by Romandini et al.

My comments and suggestions for the authors are the following:

- English language needs to be revised and improved.

Response: the text has been reviewed and corrected by a skilled English-speaking scientist of our Institution.

- Line 41: Among the main reasons for suboptimal lipid control, the authors should also mention a bad compliance to pharmacological treatment.

Response: We thank the reviewer for having raised an important point that we had inadvertently omitted. In fact, some of the new therapeutic alternatives may actually resolve or at least significantly reduce the problem of treatment compliance, such as fixed-dose statin-ezetimibe or bempedoic acid-ezetimibe combinations (which may reduce pill burden) or PCSK-9 MAbs or siRNA, which may be administered every other week, every other month or even every 3-6 months (allowing to plan and schedule the treatment in a more organized and rememberable way).  Accordingly, in the manuscript we included this further aspect among the unmet needs that new approaches and drugs may help to solve (now line 53) and we developed it below in the text (now lines 445-460).

- Throughout the manuscript, not all of the authors' claims are supported by references. For example, a reference should be added at line 106, 109, 110.

Response: in the studies where BA was added to background lipid-lowering therapy, the groups on background statin plus ezetimibe were relatively small and the data about the specific LDL-C lowering response to the addition of BA was not reported separately. Accordingly, not having this specific data, we changed the sentence:

“adding BA may be enough to approach or even reach the LDL-C target in patients with LDL-C levels very near their target on background treatment with a tolerated low-dose statin plus ezetimibe” (lines 104-106)

With…

“adding BA may be enough to approach or even reach the LDL-C target in patients with LDL-C levels very near their target on the highest tolerated doses of background SCLT” (now lines 142 – 145)

To support this notion, we added a further citation which shows a mean 23.5% additive LDL-C reduction with BA in patients on background SCLT (PMID = 29910030).

In addition, we added a new citation regarding the interactions between bempedoic acid and statins reported in the Summary of Product Characteristics (now included among the references)

- In the manuscript, all abbreviations should be specified at their first occurrence.

Response: we thank the reviewer for having noticed this imprecision. The text has been checked and all abbreviation properly specified.

- In the manuscript, the authors should also refer to findings from meta-analyses summarizing evidence in the field. In particular, they should refer to doi: 10.1371/journal.pmed.1003121,doi: 10.1007/s40264-020-00931-6, doi: 10.1161/JAHA.122.025551,doi: 10.1097/MD.0000000000031199, doi:10.1016/j.ahjo.2022.100127, doi: 10.1016/j.atherosclerosis.2020.09.021. 

Response: most of the meta-analyses suggested have properly been added. Moreover, we added the following sentence (now lines 159-160). to include one of the citations suggested:

“BA should be indicated with caution in patients with a history of hyperuricemia and/or gout (as it may increase uric acid levels and induce gout attacks)”

Regarding the paper “Additive effects of ezetimibe, evolocumab, and alirocumab on plaque burden and lipid content as assessed by intravascular ultrasound: A PRISMA-compliant meta-analysis (DOI. 10.1097/MD.0000000000031199)”, although interesting, it refers to the effects of LDL-C lowering treatments on plaque morphology which is beyond the scope of our paper and so we prefer not to include it

- Table 1 - What about mipomersen?

Response: The U.S. Food and Drug Administration (FDA) withdrew mipomersen in 2019 from the market due to its hepatotoxicity (see PMID: 35221289). In addition, mipomersen was not approved by the European Medicines Agency. For these reasons we decided not to include this compound in our “point of view paper”. Of course, if the reviewer believes that also this aspect is necessary we are ready to add a proper paragraph.

- Table 1 - References should be added in the table.

Response: Modified accordingly.

- In the manuscript, the authors should include a "methods" section detailing how literature searching process was performed.

Response: Modified accordingly (now lines 94 – 100).

- In the Conclusions, the authors should specify that this is a state-of-the-art review (and not a systematic review) and even the other limitations associated to their not systematically approach. In effect, I got the impression that the authors only reported the information and sources that supported their thesis. Readers should be properly informed of this.

Response: we agree that in this state-of-the-art paper, studies in the literature not cited in the manuscript may confute or refute our perspective. We added a sentence to openly recognize this limit of the work, early in the manuscript, (now lines 58-67).

We thank very much the reviewer for the valuable suggestions.

Reviewer 2 Report

Dear Dr

Concerning the manuscript entitled “Atherogenic dyslipidaemias: unmet needs and how the new available or upcoming drugs might fulfill them” which was submitted to the Pharmaceuticals journal by Romandini et al., there are some comments for the manuscript improvement included as followings:

1-      Title:

Please rewrite the title to be as “Atherogenic dyslipidaemias: unmet needs and the therapeutic potential of emerging and novel approaches and drugs

2-      Abstract

The writing of sentences is appropriate. Please add one or two sentences about main findings of the study. It seems that the number of keywords are more than standard. Instead of including name of all drugs, keywords can be substituted with “Atherogenic dyslipidemias, lipid-modifying drugs”, Cardiovascular disorders, Drug intolerance”.

3-      Introduction

The introduction section is appropriate. The references have been properly used.

4-      Manuscript body sections

Addition of one or more figures demonstrating mechanisms of function and text findings can make it more understandable in a shorter time for readers instead of studying a whole text in a review paper.

5-      Conclusion

It is suitable, however inclusion of main drugs examples in the conclusion will make the findings of the manuscript more appropriate and understandable.

6-      References

There is lacking of studies from 2001-2004 and 2013 in the references. If needed, these studies can be added to make the literature more powerful.

With best regards

Author Response

REVIEWER 2

Dear Dr

Concerning the manuscript entitled “Atherogenic dyslipidaemias: unmet needs and how the new available or upcoming drugs might fulfill them” which was submitted to the Pharmaceuticals journal by Romandini et al., there are some comments for the manuscript improvement included as followings:

1-      Title:

Please rewrite the title to be as “Atherogenic dyslipidaemias: unmet needs and the therapeutic potential of emerging and novel approaches and drugs

Response: Modified accordingly.

2-      Abstract

The writing of sentences is appropriate. Please add one or two sentences about main findings of the study. It seems that the number of keywords are more than standard. Instead of including name of all drugs, keywords can be substituted with “Atherogenic dyslipidemias, lipid-modifying drugs”, Cardiovascular disorders, Drug intolerance”.

Response: Modified accordingly

3-      Introduction

The introduction section is appropriate. The references have been properly used.

Response: we thank the reviewer for the positive comment.

4-      Manuscript body sections

Addition of one or more figures demonstrating mechanisms of function and text findings can make it more understandable in a shorter time for readers instead of studying a whole text in a review paper.

Response: Modified accordingly. A figure (Figure 1) briefly describing the mechanisms of action of the different new lipid-modifying compounds has been added, as the reviewer proposed.

5-      Conclusion

It is suitable, however inclusion of main drugs examples in the conclusion will make the findings of the manuscript more appropriate and understandable.

Response:  Modified accordingly

6-      References

There is lacking of studies from 2001-2004 and 2013 in the references. If needed, these studies can be added to make the literature more powerful.

Response: we did not find references regarding the new lipid-modifying compounds published in the years indicated, but if the reviewer may kindly identify the citations, we are ready to include them in the manuscript.

With best regards

Response: we thank very much the reviewer for the valuable suggestions

Round 2

Reviewer 1 Report

Dear Editor,

I carefully read the revised version of the manuscript, that is significantly improved in comparison with the original version. I warmly recommend its publication in the Journal.